# Risk of peripheral artery occlusive disease in patients with lower leg fracture who received fixation and non-fixation treatments: A population cohort study

**Pin-Keng Shih[1,2], Jian-Xun Chen[1,2], Mei-Chen Lin[1,3], Shih-Chi Wu[1,4]***

**1** School of medicine, China Medical University, Taichung, Taiwan, **2** Department of Surgery, China Medical University Hospital, Taichung, Taiwan, **3** Management Office for Health Data, China Medical University and Hospital, Taichung, Taiwan, **4** Trauma and Emergency Center, China Medical University Hospital, Taichung, Taiwan

* rw114@mail.cmuh.org.tw

## Abstract

### Background

The risk of peripheral artery occlusive disease (PAOD) in patients with lower leg fracture who underwent fixation procedures is not yet completely understood. Therefore, the current study aimed to examine the risk of subsequent PAOD in patients with lower leg fracture who received fixation and non-fixation treatments.

### Methods

We included 6538 patients with lower leg fracture who received non-fixation treatment and a matched cohort comprising 26152 patients who received fixation treatment from the National Health Insurance Database. Patients were frequency matched according to age, sex, and index year. The incidence and risk of PAOD in patients with lower leg fracture who received fixation and non-fixation treatments were evaluated via the stratification of different characteristics and comorbidities.

### Results

Non-fixation treatment, male sex, older age ($\geq$ 50 years old), diabetes mellitus, and gout were associated with a significantly higher risk of lower extremity PAOD compare to each comparison group, respectively. Moreover, there was a significant correlation between fixation treatment and a lower risk of lower extremity PAOD in women (adjusted hazard ratio [aHR] = 0.58, 95% confidence interval [CI] = 0.38–0.90), women aged > 50 years (aHR = 0.61, 95% CI = 0.38–0.96), and patients with coronary artery disease (aHR = 0.43, 95% CI = (0.23–0.81). Further, patients with fixation treatment had a significantly lower risk of lower extremity PAOD within 2 years after trauma (aHR = 0.57, 95% CI = 0.34–0.97). The Kaplan–Meier analysis showed that the cumulative incidence of PAOD was significantly

**Data Availability Statement:** This study used third party inpatient claims data from the Taiwan National Health Insurance Research Database (NHIRD) website (https://nhird.nhri.org.tw/en/

Data_Files.html). This database contains detailed medical histories of the hospitalized enrollees in Taiwan. Data are available upon request from Taiwan Ministry of Health and Welfare (TMHW) via email (nhird@nhri.edu.tw) for researchers who meet the criteria (https://nhird.nhri.org.tw/en/Data_Protection.html) for access to confidential data. The authors confirm that others would be able to access these data in the same manner as the authors. The authors also confirm that they did not have any special access privileges that others would not have.

**Funding:** This study is supported in part by Ministry of Health and Welfare, Taiwan (109-TDU-B-212-114004), MOST Clinical Trial Consortium for Stroke (MOST 108-2321-B-039-003), China Medical University Hospital (DMR-111-048). The funders had no role in study design, data collection and analysis, decision to publish, or preparation of the manuscript.

**Competing interests:** The authors have declared that no competing interests exist.

higher in the non-fixation treatment group than in the fixation treatment group at the end of the 10-year follow-up period (log-rank test: P = 0.022).

## Conclusion

Patients with lower leg fracture who received non-fixation treatment had a significantly higher risk of PAOD than those who received fixation treatment. Moreover, the risk of PAOD was higher in women aged > 50 years, as well as in coronary artery disease patients who received non-fixation treatment than in those who received fixation treatment. Therefore, regular assessment of vessel patency are recommended for these patients. Nevertheless, further studies must be conducted to validate the results of our study.

## Introduction

The prevalence of peripheral artery occlusive disease (PAOD) is about 10% in the population aged > 60 years. The risk factors may include smoking, physical inactivity, overweight, hypertension, hyperlipidemia, and hyperglycemia [1]. The signs and symptoms are intermittent claudication, ischemic pain at rest, skin ulcer, and gangrene [2]. In addition to open or endovascular revascularization, the current treatments include antiplatelet or anticoagulant therapy [3].

Lower leg fracture is managed with fixation and non-fixation procedures. In patients with stable tibial fractures and well-aligned tibial component, fracture can be successfully treated with closed reduction and cast immobilization as well as cautious maintenance of alignment during follow-up [4]. In patients with comminuted tibial fracture who present with component instability or misalignment, open reduction with internal or external fixation is commonly indicated to restore alignment. Moreover, it may be associated with a shorter hospital stay, faster pain relief, and lower risk of complications [5].

A previous study showed that 1.04% of patients with lower extremity blunt fracture present with acute artery injury [6]. Further, patients with open tibial fracture commonly develop chronic PAOD [7]. Hence, emerging evidence has shown that PAOD is associated with lower leg fracture [6, 7]. Although the mechanism underlying PAOD is not fully elucidated, it might be caused by mechanical or biochemical factors [8]. Some case studies have revealed that patients with comminuted fracture have a greater risk of PAOD, which might be attributed to a high energy impact and vessel damage [8, 9].

Therefore, the current study aimed to examine the risk of subsequent PAOD in patients with lower leg fracture who received fixation and non-fixation treatments.

## Materials and methods

### Data source

Taiwan established the National Health Insurance Program, which enrolled nearly 99% of residents. Since 1995, comprehensive health claimed data have been stored in the National Health Insurance Research Database (NHIRD). This database contains information about prescription, treatment, medical costs, and other medical services for both inpatients and outpatients. The current research used data from the population-based hospitalization database. To protect the privacy of each participant, the identification numbers were already encrypted before data were released by the government. The diagnoses in the Taiwan NHI are defined according to

the International Classification of Disease, Ninth Revision, Clinical Modification (ICD-9-CM). This study was approved by the research ethics committee of China Medical University and Hospital in Taiwan [CMUH-104-REC2-115-CR6].

## Definition of fixation treatment

Fixation treatment was defined as internal or external fixation and non-fixation treatment as short leg casting.

## Study population

To validate the association between fixation treatment and the risk of arterial embolism and thrombosis in the lower extremity (lower extremity PAOD) in patients with tibial and fibula fracture, we identified patients with newly diagnosed tibial and fibula fracture (ICD-9-CM: 823) from 2000 to 2012 who received internal or external fixation treatment (ICD-9-OP: 78.17, 78.57, 79.16, and 79.36). These patients were then assigned to the fixation treatment group, and those who did not receive internal or external fixation treatment to the non-fixation treatment group (ICD-9-OP: 79.06, and 79.26). The index date was set as the fixation date. Patients diagnosed with lower extremity PAOD (ICD-9-CM: 444.22) before the index date or those aged < 18 years (calculated until the index date) were excluded from the study.

## Baseline comorbidity and outcome

Comorbidities were important confounding factors, and those observed before the index date included diabetes (ICD-9-CM: 250) [10], hypertension (ICD-9-CM: 401–405) [11], gout (ICD-9-CM: 274) [12], hyperlipidemia (ICD-9-CM: 272) [13], mental disorder (ICD-9-CM: 290, 294–298, 311) [14], stroke (ICD-9-CM: 430–438) [15], Parkinson disease (ICD-9-CM: 332) [16], coronary artery disease (ICD-9-CM: 410–414) [17], heart failure (ICD-9-CM: 428) [17], chronic obstructive pulmonary disease (COPD) (ICD-9-CM: 491, 492, 496) [18], liver disease (ICD-9-CM: 571, 572) [19], and end-stage renal disease (ESRD) (ICD-9-CM: 585) [20]

All participants were followed-up until the occurrence of lower extremity PAOD, death, and withdrawal from the NHIRD program or until December 31, 2013.

## Statistical analysis

We performed propensity score matching. Then, to assess probabilities, which were used to assign four fixation-treated patients to each non-treated participant, the propensity score was calculated via a logistic regression analysis. The matched variables included sex, age, index year of fixation treatment, diabetes, hypertension, and gout.

The baseline demographic characteristics and comorbidities were identified. The difference between the two cohorts was assessed using standardized mean difference (SMD). Results showed a negligible difference, with a SMD of $\leq 0.1$ [21]. The incidence of lower extremity PAOD was calculated as the number of newly occurred lower extremity PAOD divided by the following period (per 10,000 person-year). The Kaplan–Meier method was applied to calculate the cumulative incidence of lower extremity PAOD, and the difference between two survival curves were assessed using the log-rank test. Furthermore, the potential risk factors of lower extremity PAOD were assessed, and stratified analyses were conducted using the Cox proportional hazard model. Results were presented as hazard ratio, adjusted hazard ratio (aHR) (for demographic factors and comorbidities), and 95% confidence interval (95% CI). All statistical analyses were performed using the SAS statistical software version 9.4 (SAS Institute Inc.,

Cary, NC). The cumulative incidence curve was plotted using the R software. A two-sided p-value of < 0.05 was considered statistically significant.

## Results

In total, 32690 participants with lower leg fracture, including 26152 with fixation treatment and 6538 with non-fixation treatment, were enrolled in this study.

In the fixation group, there were 10913 (41.7%) women and 15239 (58.3%) men. Moreover, 4736 (18.1%) patients were aged 20–29 years; 3732 (14.3%) patients were aged 30–39 years; 4644 (17.8%) patients were aged 40–49 years; 4747 (18.2%) patients were aged 50–59 years; 3944 (15.1%) patients were aged 60–69 years; 3230 (12.4%) patients were aged 70–79 years, and 1119 (4.3%) ≥ 80 years.

In the non-fixation group, there were 2770 (42.4%) women and 3768 (57.6%) men. Further, 1175 (18.0%) patients were aged 20–29 years; 948 (14.5%) patients were aged 30–39 years; 1196 (18.3%) patients were aged 40–49 years; 1157 (17.7%) patients were aged 50–59 years; 941 (14.4%) patients were aged 60–69 years; 744 (11.4%) patients were aged 70–79 years, and 377 (5.8%) ≥ 80 years.

There were no significant differences in terms of sex, age, and baseline comorbidities between the two groups (SMD of < 0.1) (Table 1).

**Table 1. Demographic characteristics and comorbidities of patients newly diagnosed fracture of tibia and fibula in Taiwan during 2000–2012.**

| Characteristics | Total | Fracture of tibia and fibula | | SMD |
|---|---|---|---|---|
| | | With internal or external fixation N = 26152 | Without internal or external fixation N = 6538 | |
| **Gender** | | | | 0.013 |
| Female | 13683 | 10913 (41.7) | 2770 (42.4) | |
| Male | 19007 | 15239 (58.3) | 3768 (57.6) | |
| **Age, mean (SD)** | | 49.8 (18.3) | 50.0 (18.6) | 0.011 |
| 20–29 | 5911 | 4736 (18.1) | 1175 (18) | |
| 30–39 | 4680 | 3732 (14.3) | 948 (14.5) | |
| 40–49 | 5840 | 4644 (17.8) | 1196 (18.3) | |
| 50–59 | 5904 | 4747 (18.2) | 1157 (17.7) | |
| 60–69 | 4885 | 3944 (15.1) | 941 (14.4) | |
| 70–79 | 3974 | 3230 (12.4) | 744 (11.4) | |
| >=80 | 1496 | 1119 (4.3) | 377 (5.8) | |
| **Baseline comorbidity** | | | | |
| Hypertension | 3181 | 2471 (9.4) | 710 (10.9) | 0.047 |
| Diabetes mellitus | 2543 | 1999 (7.6) | 544 (8.3) | 0.025 |
| Gout | 547 | 412 (1.6) | 135 (2.1) | 0.037 |
| Hyperlipidemia | 948 | 721 (2.8) | 227 (3.5) | 0.041 |
| Mental disorder | 648 | 495 (1.9) | 153 (2.3) | 0.031 |
| Stroke | 1271 | 992 (3.8) | 279 (4.3) | 0.024 |
| Parkinson disease | 129 | 106 (0.4) | 23 (0.4) | 0.009 |
| Coronary artery disease | 1474 | 1141 (4.4) | 333 (5.1) | 0.034 |
| Heart failure | 546 | 425 (1.6) | 121 (1.9) | 0.017 |
| COPD | 846 | 663 (2.5) | 183 (2.8) | 0.016 |
| Liver disease | 1617 | 1243 (4.8) | 374 (5.7) | 0.043 |
| ESRD | 170 | 132 (0.5) | 38 (0.6) | 0.010 |

§A standardized mean difference of ≤0.1 indicates a negligible difference

Table 2 depicts the incidence and risk factors of lower extremity PAOD. After adjusting for age, sex, and all comorbidities, male sex (aHR = 1.49, 95% CI = 1.14–1.96), older age (≥50 years, aHR = 4.03, 95% CI = 2.73–5.95), diabetes mellitus (aHR = 4.06, 95% CI = (2.95–5.59), gout (aHR = 2.08, 95% CI = (1.25–3.46), coronary artery disease (aHR = 1.63, 95% CI = (1.10–2.40), and ESRD (aHR = 4.88, 95% CI = (2.39–10.00) were found to be associated with a significantly higher risk of lower extremity PAOD compare to each comparison group, respectively. The fixation treatment group (aHR = 0.71, 95% CI = (0.53–0.95) had a significantly lower risk of PAOD than the non-fixation treatment group.

Table 3 shows the findings of the univariate and multivariate stratified analyses. Results showed a significant association between fixation treatment and a reduced risk of lower extremity PAOD among women (aHR = 0.58, 95% CI = (0.38–0.90) and women aged > 50 years (aHR = 0.61, 95% CI = (0.38–0.96), and patients with coronary artery disease (aHR = 0.43, 95% CI = (0.23–0.81).

After stratifying follow-up years into < 2, 2–5, and > 5 years, patients with fixation treatment had a significantly lower risk of lower extremity PAOD in the 2-year follow-up (aHR = 0.57, 95% CI = 0.34–0.97) (Table 4).

Fig 1 shows that the fixation treatment group had a significantly lower cumulative incidence of lower extremity PAOD than the non-fixation treatment group in the 10-year follow-up (log-rank test, P = 0.022).

**Table 2. Cox model measured hazard ratio and 95% confidence intervals of arterial embolism and thrombosis of lower extremity associated with and without internal or external fixation among Fracture of tibia and fibula patients.**

| Characteristics | Event | Crude | | Adjusted | |
|---|---|---|---|---|---|
| | (n = 233) | HR (95% CI) | p value | HR (95% CI) | p value |
| **Internal or external fixation** | | | | | |
| No | 59 | Ref. | | Ref. | |
| Yes | 174 | 0.71 (0.53–0.95) | 0.023 | 0.71 (0.53–0.95) | 0.023 |
| **Gender** | | | | | |
| Female | 97 | Ref. | | Ref. | |
| Male | 136 | 1.00 (0.77–1.30) | 1.000 | 1.49 (1.14–1.96) | 0.004 |
| **Age at baseline** | | | | | |
| <50 | 35 | Ref. | | Ref. | |
| ≥50 | 198 | 7.73 (5.08–10.43) | <0.001 | 4.03 (2.73–5.95) | <0.001 |
| **Baseline comorbidity** | | | | | |
| Hypertension | 74 | 6.76 (5.11–8.94) | <0.001 | 1.13 (0.79–1.62) | 0.500 |
| Diabetes mellitus | 85 | 10.68 (8.15–14.00) | <0.001 | 4.06 (2.95–5.59) | <0.001 |
| Gout | 19 | 8.22 (5.13–13.17) | <0.001 | 2.08 (1.25–3.46) | 0.005 |
| Hyperlipidemia | 28 | 6.22 (4.19–9.25) | <0.001 | 1.28 (0.81–2.01) | 0.294 |
| Mental disorder | 5 | 1.54 (0.63–3.73) | 0.3419 | 0.66 (0.27–1.63) | 0.369 |
| Stroke | 30 | 5.80 (3.94–8.53) | <0.001 | 1.09 (0.71–1.68) | 0.685 |
| Parkinson disease | 2 | 3.88 (0.96–15.61) | 0.057 | 0.99 (0.24–4.06) | 0.985 |
| Coronary artery disease | 45 | 7.31 (5.28–10.14) | <0.001 | 1.63 (1.10–2.40) | 0.014 |
| Heart failure | 16 | 7.89 (4.73–13.14) | <0.001 | 1.13 (0.64–2.00) | 0.667 |
| COPD | 17 | 4.52 (2.76–7.42) | <0.001 | 0.83 (0.49–1.40) | 0.489 |
| Liver disease | 33 | 4.33 (2.99–6.27) | <0.001 | 1.39 (0.94–2.07) | 0.102 |
| ESRD | 9 | 16.61 (8.49–32.50) | <0.001 | 4.88 (2.39–10.00) | <0.001 |

*Abbreviation: HR, hazard ratio; CI, confidence interval.

*Adjusted HR: adjusted for gender, age, and comorbidities in Cox proportional hazards regression.

**Table 3. Incidence rates, hazard ratio and confidence intervals of arterial embolism and thrombosis of lower extremity in different stratification.**

| Variables | With internal or external fixation | | | Without internal or external fixation | | | Compared to without internal or external fixation | | | |
|---|---|---|---|---|---|---|---|---|---|---|
| | n = 26152 | | | n = 6538 | | | Crude HR | p-value | Adjusted HR | p-value |
| | Event | Person years | IR | Event | Person years | IR | (95% CI) | | (95% CI) | |
| **Gender** | | | | | | | | | | |
| Female | 67 | 86482 | 7.75 | 30 | 20945 | 14.32 | 0.54 (0.35–0.83) | 0.005 | 0.58 (0.38–0.90) | 0.014 |
| <50 | 5 | 39246 | 1.27 | 3 | 9706 | 3.09 | 0.41 (0.10–1.71) | 0.222 | 0.53 (0.11–2.50) | 0.420 |
| ≥50 | 62 | 47237 | 13.13 | 27 | 11238 | 24.02 | 0.55 (0.35–0.86) | 0.009 | 0.61 (0.38–0.96) | 0.032 |
| Male | 107 | 120665 | 8.87 | 29 | 28983 | 10.01 | 0.88 (0.59–1.33) | 0.556 | 0.83 (0.55–1.25) | 0.825 |
| **Age at baseline** | | | | | | | | | | |
| <50 | 23 | 113753 | 2.02 | 12 | 28292 | 4.24 | 0.48 (0.24–0.95) | 0.037 | 0.53 (0.26–1.07) | 0.078 |
| ≥50 | 151 | 93395 | 16.17 | 47 | 21636 | 21.72 | 0.74 (0.53–1.03) | 0.071 | 0.74 (0.53–1.03) | 0.073 |
| **Baseline comorbidity** | | | | | | | | | | |
| Hypertension | 55 | 13928 | 39.49 | 19 | 3643 | 52.16 | 0.74 (0.44–1.25) | 0.267 | 0.69 (0.41–1.16) | 0.161 |
| Diabetes mellitus | 62 | 11121 | 55.75 | 23 | 2929 | 78.54 | 0.71 (0.44–1.15) | 0.163 | 0.70 (0.43–1.13) | 0.146 |
| Gout | 13 | 2164 | 60.08 | 6 | 705 | 85.07 | 0.72 (0.27–1.90) | 0.507 | 0.52 (0.18–1.52) | 0.234 |
| Hyperlipidemia | 20 | 4321 | 46.28 | 8 | 1382 | 57.91 | 0.80 (0.35–1.83) | 0.602 | 0.69 (0.30–1.58) | 0.373 |
| Mental disorder | 3 | 2919 | 10.28 | 2 | 827 | 24.20 | 0.43 (0.07–2.60) | 0.360 | 0.50 (0.05–4.89) | 0.551 |
| Stroke | 23 | 5274 | 43.61 | 7 | 1403 | 49.89 | 0.88 (0.38–2.06) | 0.772 | 0.81 (0.34–1.91) | 0.622 |
| Parkinson disease | 1 | 493 | 20.28 | 1 | 104 | 96.07 | 0.22 (0.01–3.52) | 0.284 | - | - |
| Coronary artery disease | 29 | 6619 | 43.81 | 16 | 1787 | 89.52 | 0.49 (0.26–0.90) | 0.021 | 0.43 (0.23–0.81) | 0.008 |
| Heart failure | 11 | 1996 | 55.12 | 5 | 449 | 111.31 | 0.49 (0.17–1.41) | 0.186 | 0.39 (0.12–1.24) | 0.109 |
| COPD | 14 | 3681 | 38.04 | 3 | 860 | 34.87 | 1.07 (0.31–3.75) | 0.911 | 0.87 (0.24–3.20) | 0.832 |
| Liver disease | 24 | 7471 | 32.13 | 9 | 2139 | 42.07 | 0.74 (0.34–1.60) | 0.443 | 0.67 (0.31–1.46) | 0.314 |
| ESRD | 6 | 518 | 115.83 | 3 | 129 | 232.35 | 0.52 (0.13–2.07) | 0.352 | 0.43 (0.06–2.86) | 0.380 |

*Abbreviation: IR, incidence rates, per 10,000 person-years; HR, hazard ratio; CI, confidence interval.

*Adjusted HR: adjusted for gender, age, and comorbidities in Cox proportional hazards regression.

## Discussion

Our study showed that female patients aged > 50 years who received fixation treatment had a significantly lower risk of PAOD than those who received non-fixation treatment. However, there was no significant difference among male patients (Table 3).

After stratification according to different follow-up periods, patients with fixation treatment had a significantly lower risk of lower extremity PAOD within 2 years after trauma than those with non-fixation treatment.

**Table 4. Incidence rates, hazard ratio and confidence intervals of arterial embolism and thrombosis of lower extremity in different follow-up stratification.**

| Variables | With internal or external fixation | | | Without internal or external fixation | | | Compared to without internal or external fixation | | | |
|---|---|---|---|---|---|---|---|---|---|---|
| | n = 26152 | | | n = 6538 | | | Crude HR | p-value | Adjusted HR | p-value |
| | Event | Person years | IR | Event | Person years | IR | (95% CI) | | (95% CI) | |
| **Follow-up years** | | | | | | | | | | |
| <2 | 45 | 50617 | 8.89 | 21 | 12442 | 16.88 | 0.53 (0.32–0.89) | 0.016 | 0.57 (0.34–0.97) | 0.036 |
| 2–5 | 36 | 66060 | 5.45 | 10 | 16065 | 6.22 | 0.88 (0.43–1.76) | 0.709 | 0.86 (0.42–1.73) | 0.663 |
| >5 | 93 | 90470 | 10.28 | 28 | 21421 | 13.07 | 0.79 (0.52–1.20) | 0.262 | 0.74 (0.48–1.13) | 0.164 |

*Abbreviation: IR, incidence rates, per 10,000 person-years; HR, hazard ratio; CI, confidence interval.

*Adjusted HR: adjusted for gender, age, and comorbidities in Cox proportional hazards regression.

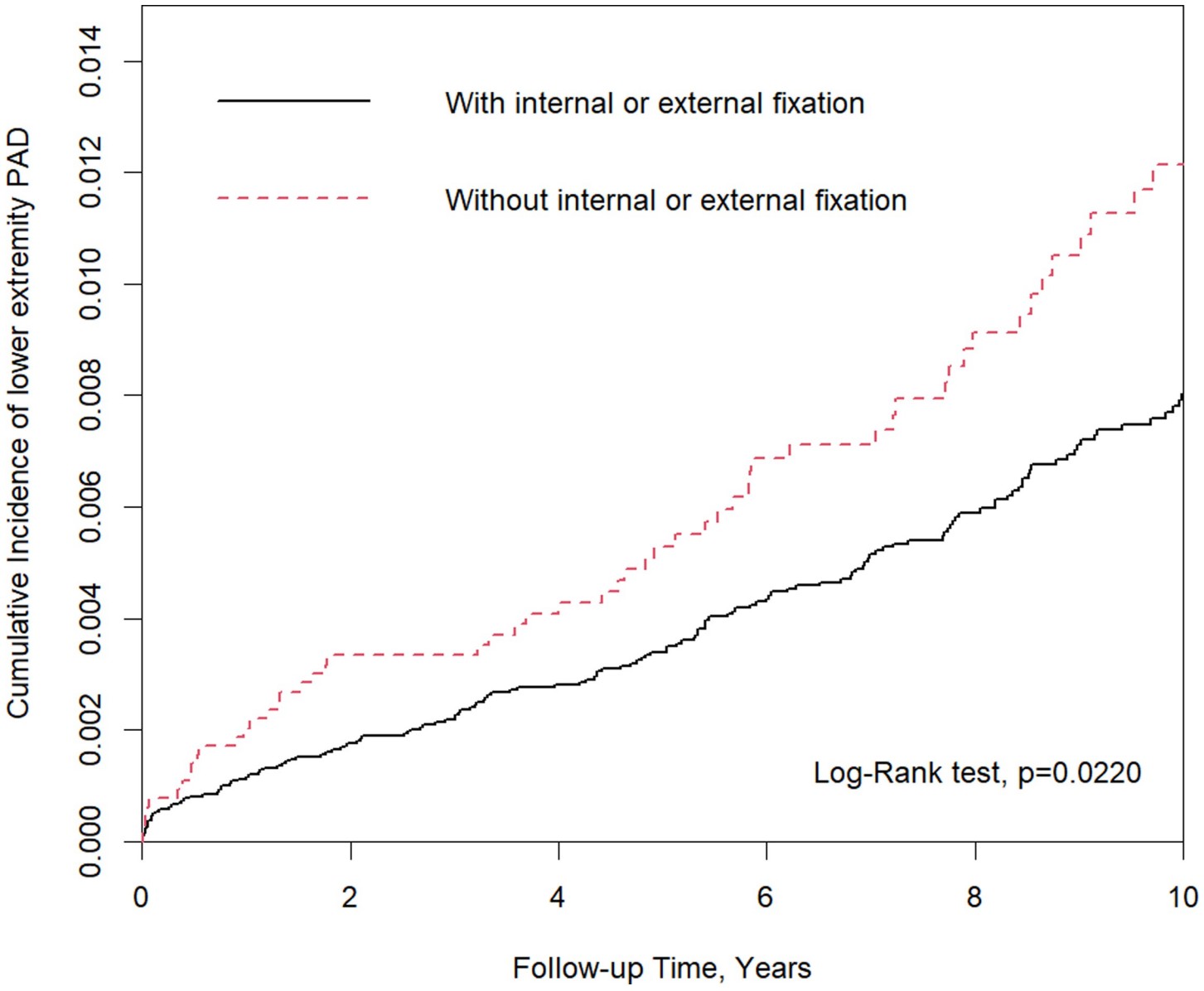

**Fig 1. Cumulative incidence curves of peripheral artery disease in patients with lower leg fracture who received fixation treatment and those who did not.**

Previous studies have shown that patients with hip trauma are at risk of PAOD [22, 23]. To prevent misinterpretation, patients with hip fracture (ICD codes: 835, 9240, and 9280) were excluded from this research. Therefore, the term lower leg fracture refers to tibial, fibular, or tibiofibular fracture.

It is rational to conclude that patients with fixation treatment had a higher risk of PAOD than those without due to a more severe tissue damage. Nevertheless, our results showed that the non-fixation treatment group had a higher risk of PAOD than the fixation treatment group (Table 2). This could be explained by the fact that patients with lower leg fracture who did not undergo fixation surgery commonly had cast immobilization, which may increase the risk of PAOD. By contrast, some studies have revealed that prolonged lower leg immobilization after fracture was associated with venous thrombosis formation [24, 25]. However,

whether this finding is true for artery compromise such as that in PAOD patients with lower leg fracture must be further investigated. Taken together, we believe that a high energy impact in lower leg comminuted fracture can not only damage bone alignment but also affect vessel patency [26, 27]. Hence, patients with this condition should receive fixation treatment [28]. Since only few studies have focused on the association between PAOD and immobilization after lower leg fracture, further investigations should be performed.

Women aged > 50 years who received non-fixation treatment had a higher risk of PAOD than those who received fixation treatment (Table 3). Moreover, they often present with menopause, which is associated with a higher risk of osteoporosis and fracture [29]. Therefore, osteoporosis can enhance the need for a longer immobilization after fracture in aging women with non-fixation treatment, thereby resulting in a higher risk for PAOD. Although osteoporosis may increase the risk of lower leg fracture, fixation may decrease the need for immobilization, which can reduce the incidence of PAOD in older female patients who underwent surgery (Table 3).

There was a significant association between fixation treatment and a reduced risk of lower extremity PAOD in patients with coronary artery disease (Table 3). It has been reported that there are close relations between coronary artery disease and PAOD [17, 30, 31]. Yet, due to limited patient number, further studies are required to identify whether coronary artery disease patients with lower leg fracture are proposed to receive fixation treatment.

Some studies have shown that chronic artery insufficiency presents as nonunion of fracture during the 2-year follow-up in patients with lower leg fracture [7, 32, 33]. In this study, compared with the fixation group, the non-fixation treatment group had a higher risk of PAOD within 2 years after trauma, and there were no differences in terms of risk after 2 years (Table 4). A time period of 2 years might be required to progressively establish collateral blood circulation after trauma. However, further studies must be performed to validate this result.

To the best of our knowledge, this is the first large-scale study with a long-term follow-up about the subsequent risk of PAOD in patients with lower leg fracture who received fixation and non-fixation treatments. In the current series, the cumulative incidence of PAOD was significantly higher in the non-fixation treatment group than in the fixation treatment group (Fig 1). In addition, women aged > 50 years, as well as coronary artery disease patients with lower leg fracture who received non-fixation treatment had a higher risk of PAOD than those who received fixation treatment (Table 3). Therefore, regular assessment of vessel patency via ultrasonography and treatment with prophylactic anticoagulation drugs are recommended for these patients.

## Limitations

With reliable diagnoses and a high rate of follow-up, the assessment of PAOD risk in patients with lower leg fracture who received fixation and non-fixation treatments is strengthened by the inclusion of a large number of patients and as well as longitudinal assessments and subgroup analyses. However, the current research had several limitations. First, data about lifestyles factors, such as smoking, dietary habits, drinking, and socioeconomic status, and genetic factors were not obtained for the adjustment of PAOD risk. Moreover, a stratified analysis of patients with simple and compound (comminuted) lower leg fracture was not performed because of databank limitation. Second, PAOD was diagnosed based on admission diagnoses and codes. Thus, an extremely accurate analysis was not performed, and there was no accessible information regarding fracture severity/grade. Similarly, owing to database limitation, it would be very difficult to differentiate and analyze the fixation methods separately for internal and external fixation, as well as measure the proportion of minimally invasive plate osteosynthesis among the internal fixations population. Third, because all data were anonymized, relevant clinical variables, such as surgical findings, imaging results, and laboratory data, were not available. Fourth,

biases could have existed due to the retrospective nature of the study. Nevertheless, data regarding lower leg fracture, surgery, and PAOD diagnosis were highly reliable.

## Conclusion

Patients with lower leg fracture who received non-fixation treatment had a significantly higher risk of PAOD than those who received fixation treatment. Moreover, the risk of PAOD was higher in women aged > 50 years, as well as in coronary artery disease patients who received non-fixation treatment than in those who received fixation treatment. Nevertheless, further studies should be conducted to validate our results.

## Author Contributions

**Conceptualization:** Pin-Keng Shih, Shih-Chi Wu.

**Data curation:** Jian-Xun Chen.

**Formal analysis:** Jian-Xun Chen, Mei-Chen Lin.

**Methodology:** Mei-Chen Lin.

**Writing – original draft:** Pin-Keng Shih, Shih-Chi Wu.

**Writing – review & editing:** Shih-Chi Wu.

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
