## [Decision Letter · Decision Letter 0]

30 Mar 2022

PONE-D-22-00911Risk of Peripheral Artery Occlusive Disease in Patients with Lower Leg Fracture Who Received Fixation and Non-fixation Treatments: A Population Cohort StudyPLOS ONE

Dear Dr. Wu,

Thank you for submitting your manuscript to PLOS ONE. After careful consideration, we feel that it has merit but does not fully meet PLOS ONE’s publication criteria as it currently stands. Therefore, we invite you to submit a revised version of the manuscript that addresses the points raised during the review process.

As you will recognize from the comments of the reviewers they raised major points of critique, especially regarding study design, presentation of data and ethical approval.

Please submit your revised manuscript within May 14 2022 11:59PM. If you will need more time than this to complete your revisions, please reply to this message or contact the journal office at plosone@plos.org. Please include the following items when submitting your revised manuscript:A rebuttal letter that responds to each point raised by the academic editor and reviewer(s). You should upload this letter as a separate file labeled 'Response to Reviewers'.A marked-up copy of your manuscript that highlights changes made to the original version. You should upload this as a separate file labeled 'Revised Manuscript with Track Changes'.An unmarked version of your revised paper without tracked changes. You should upload this as a separate file labeled 'Manuscript'.

We look forward to receiving your revised manuscript.

Kind regards,

Rudolf Kirchmair

Academic Editor

PLOS ONE

Journal Requirements:

“This study is supported in part by Ministry of Health and Welfare, Taiwan (109-TDU-B-212-114004), MOST Clinical Trial Consortium for Stroke (MOST 108-2321-B-039-003).

We note that you have provided additional information within the Funding Section that is not currently declared in your Funding Statement. Please note that funding information should not appear in the Funding section or other areas of your manuscript. We will only publish funding information present in the Funding Statement section of the online submission form.

Additional Editor Comments (if provided):

Reviewers' comments:

Reviewer's Responses to Questions

**Comments to the Author**

1. Is the manuscript technically sound, and do the data support the conclusions?

Reviewer #1: Yes

Reviewer #2: No

2. Has the statistical analysis been performed appropriately and rigorously? 

Reviewer #1: Yes

Reviewer #2: No

3. Have the authors made all data underlying the findings in their manuscript fully available?

Reviewer #1: Yes

Reviewer #2: No

4. Is the manuscript presented in an intelligible fashion and written in standard English?

Reviewer #1: Yes

Reviewer #2: No

5. Review Comments to the Author

Reviewer #1: I think it is a very significant paper to improve the quality of trauma care for functional and long-term prognosis. Please correct the following two points.

Major revision

Soft tissue damage and fixation period are thought to influence vessel patency, but if this is the case, it would be better to analyze the fixation methods separately for internal and external fixation. Also, what was the proportion of minimally invasive plate osteosynthesis among the internal fixations? It would be good to at least mention that in the limitation.

Minor revision

In the preface to “Results”, the numerical notation is separated by every three digits, but not thereafter. Please unify either.

Reviewer #2: Comments to authors:

1.

In Table 1, the authors need to include more history of diseases or coexisting medical conditions, such as hyperlipidemia, mental disorders, stroke, Parkinson’s disease, ischemic heart disease, heart failure, COPD, liver cirrhosis, and renal dialysis. In addition, the use of medical care service also should be considered in this study.

2.

In Table 1, the age should be divided as 20-29, 30-39, 40-49, 50-59, 60-69, 70-79, >=80.

3.

The immortal time bias is very important to this study and the authors need to consider this point in the study design and statistical analysis.

4.

The IRB number seems very old. The authors need to provide the real ethical approval from the Intuitional Review Board.

6. PLOS authors have the option to publish the peer review history of their article (what does this mean?). If published, this will include your full peer review and any attached files.

Reviewer #1: No

Reviewer #2: No

---

## [Author Response · Author response to Decision Letter 0]

2 Jun 2022

Response to Reviewers’ comments

PONE-D-22-00911

Risk of Peripheral Artery Occlusive Disease in Patients with Lower Leg Fracture Who Received Fixation and Non-fixation Treatments: A Population Cohort Study

PLOS ONE

Dear Prof. Rudolf Kirchmair and reviewers of PLOS ONE,

Thank you very much for the Letter with the Reviewers’ comments about our manuscript. We greatly appreciate reviewers’ helpful suggestions and the opportunity to revise our manuscript. Efforts have been made to address each comment/concern.

Please find attached revision of the manuscript and see below for our response to the reviewers’ comments.

We look forward to your response.

Yours Sincerely,

Dr. Shih-Chi Wu

Trauma and Emergency Center

China Medical University Hospital

Taichung, Taiwan

Tel: 886-4-22052121 ext 2933

E-mail: rw114@mail.cmuh.org.tw

Response to Reviewers’ comments

Reviewer #1: I think it is a very significant paper to improve the quality of trauma care for functional and long-term prognosis. Please correct the following two points.

Major revision

Soft tissue damage and fixation period are thought to influence vessel patency, but if this is the case, it would be better to analyze the fixation methods separately for internal and external fixation.

Response: Thank you very much. We completely agree your comment and greatly appreciate this very crucial point. Yet, owing to database limitation, it would be very difficult to differentiate and analyze the fixation methods separately for internal and external fixation. Therefore, we have added related description in the section of “Limitation” 

Also, what was the proportion of minimally invasive plate osteosynthesis among the internal fixations? It would be good to at least mention that in the limitation.

Response: Similar to the above comment. It would be very difficult to measure the proportion of minimally invasive plate osteosynthesis among the internal fixations population. Thank you very much for this point, we have added related description in the section of “Limitation”

Minor revision

In the preface to “Results”, the numerical notation is separated by every three digits, but not thereafter. Please unify either.

Response: Thank you very much, we have made correction. 

Reviewer #2: 

1. In Table 1, the authors need to include more history of diseases or coexisting medical conditions, such as hyperlipidemia, mental disorders, stroke, Parkinson’s disease, ischemic heart disease, heart failure, COPD, liver cirrhosis, and renal dialysis. 

Response: Thank you very much for this point. We greatly appreciate this very crucial comment. We have performed a new analysis following your comment, and added references in the section of “Materials and Methods’. Please refer to the newly revised Table 1-4. 

Additionally, we found that there was a significant association between fixation treatment and a reduced risk of lower extremity PAOD in patients with coronary artery disease (Table 3). Therefore, we have added description in the section of “Discussion”

Table 1. Demographic characteristics and comorbidities of patients newly diagnosed Fracture of tibia and fibula in Taiwan during 2000-2012

Characteristics Total Fracture of tibia and fibula SMD

 With internal or external fixation

N=26152 Without internal or external fixation

N=6538 

Gender 0.013

Female 13683 10913 (41.7) 2770 (42.4) 

Male 19007 15239 (58.3) 3768 (57.6) 

Age, mean (SD) 49.8 (18.3) 50.0 (18.6) 0.011

20-29 5911 4736 (18.1) 1175 (18) 

30-39 4680 3732 (14.3) 948 (14.5) 

40-49 5840 4644 (17.8) 1196 (18.3) 

50-59 5904 4747 (18.2) 1157 (17.7) 

60-69 4885 3944 (15.1) 941 (14.4) 

70-79 3974 3230 (12.4) 744 (11.4) 

>=80 1496 1119 (4.3) 377 (5.8) 

Baseline comorbidity 

Hypertension 3181 2471 (9.4) 710 (10.9) 0.047

Diabetes mellitus 2543 1999 (7.6) 544 (8.3) 0.025

Gout 547 412 (1.6) 135 (2.1) 0.037

Hyperlipidemia 948 721 (2.8) 227 (3.5) 0.041

Mental disorder 648 495 (1.9) 153 (2.3) 0.031

Stroke 1271 992 (3.8) 279 (4.3) 0.024

Parkinson disease 129 106 (0.4) 23 (0.4) 0.009

Coronary artery disease 1474 1141 (4.4) 333 (5.1) 0.034

Heart failure 546 425 (1.6) 121 (1.9) 0.017

COPD 846 663 (2.5) 183 (2.8) 0.016

Liver disease 1617 1243 (4.8) 374 (5.7) 0.043

ESRD 170 132 (0.5) 38 (0.6) 0.010

§A standardized mean difference of ≤0.1 indicates a negligible difference 

In addition, the use of medical care service also should be considered in this study.

Response: Many thanks for this point. The use of medical care service could be considered equal in Taiwan. Please refer to the following description.

“Taiwan has established the National Health Insurance Program. Inhabitants of Taiwan are obligated to join this health program, which enrolled nearly 99% of residents. Therefore, patients in Taiwan shared almost the same medical care service. 

In addition, since 1995, comprehensive health claimed data have been stored in the National Health Insurance Research Database (NHIRD). The NHIRD covered more than 99% of the population in Taiwan. In addition, there were validation studies on this registry which showed that NHIRD is a large, powerful data source for biomedical research.” 

References:

1. Hsing AW, Ioannidis JP. Nationwide Population Science: Lessons From the Taiwan National Health Insurance Research Database. JAMA Intern Med. 2015 Sep; 175(9):1527-9.

2. Lin LY, Warren-Gash C, Smeeth L, et al. Data resource profile: the National Health Insurance Research Database (NHIRD). Epidemiol Health. 2018;40:e2018062.

3. Hsieh CY, Su CC, Shao SC, et al. Taiwan's National Health Insurance Research Database: past and future. Clin Epidemiol. 2019; 11:349–358.

2. In Table 1, the age should be divided as 20-29, 30-39, 40-49, 50-59, 60-69, 70-79, >=80.

Response: Thanks a lot for this comment. We have performed a new analysis following your comment. Please refer to the newly revised Table 1. 

Table 1. Demographic characteristics and comorbidities of patients newly diagnosed Fracture of tibia and fibula in Taiwan during 2000-2012

Characteristics Total Fracture of tibia and fibula SMD

 With internal or external fixation

N=26152 Without internal or external fixation

N=6538 

Gender 0.013

Female 13683 10913 (41.7) 2770 (42.4) 

Male 19007 15239 (58.3) 3768 (57.6) 

Age, mean (SD) 49.8 (18.3) 50.0 (18.6) 0.011

20-29 5911 4736 (18.1) 1175 (18) 

30-39 4680 3732 (14.3) 948 (14.5) 

40-49 5840 4644 (17.8) 1196 (18.3) 

50-59 5904 4747 (18.2) 1157 (17.7) 

60-69 4885 3944 (15.1) 941 (14.4) 

70-79 3974 3230 (12.4) 744 (11.4) 

>=80 1496 1119 (4.3) 377 (5.8) 

Occupation 0.013

Office workers 15109 12092 (46.2) 3017 (46.1) 

Manual workers 13833 11043 (42.2) 2790 (42.7) 

Others 3748 3017 (11.5) 731 (11.2) 

Baseline comorbidity 

Hypertension 3181 2471 (9.4) 710 (10.9) 0.047

Diabetes mellitus 2543 1999 (7.6) 544 (8.3) 0.025

Gout 547 412 (1.6) 135 (2.1) 0.037

Hyperlipidemia 948 721 (2.8) 227 (3.5) 0.041

Mental disorder 648 495 (1.9) 153 (2.3) 0.031

Stroke 1271 992 (3.8) 279 (4.3) 0.024

Parkinson disease 129 106 (0.4) 23 (0.4) 0.009

Coronary artery disease 1474 1141 (4.4) 333 (5.1) 0.034

Heart failure 546 425 (1.6) 121 (1.9) 0.017

COPD 846 663 (2.5) 183 (2.8) 0.016

Liver disease 1617 1243 (4.8) 374 (5.7) 0.043

ESRD 170 132 (0.5) 38 (0.6) 0.010

§A standardized mean difference of ≤0.1 indicates a negligible difference 

3. The immortal time bias is very important to this study and the authors need to consider this point in the study design and statistical analysis.

Response: Many thanks for this important comment. 

All the patient data in this study came from the population-based hospitalization database, whereas the diagnosis (tibia or fibular fracture) was coded immediately after admission, and the index date initiated in the treatment and non-treatment group. Due to the policy of Diagnosis Related Groups (DRG) in extremity fracture in our National Health Insurance (i.e. package fee for single case with extremity fracture), patients in the treatment group (internal or external fixation) received surgery soon after diagnosis. Therefore, the duration from established diagnosis to treatment was short (within very few days) while the immortal time bias could be neglected. 

4. The IRB number seems very old. The authors need to provide the real ethical approval from the Intuitional Review Board.

Response: Thank you very much for this important point. 

Please refer to the following IRB certificate.

---

## [Decision Letter · Decision Letter 1]

13 Jul 2022

Risk of Peripheral Artery Occlusive Disease in Patients with Lower Leg Fracture Who Received Fixation and Non-fixation Treatments: A Population Cohort Study

PONE-D-22-00911R1

Dear Dr. Wu,

We’re pleased to inform you that your manuscript has been judged scientifically suitable for publication and will be formally accepted for publication once it meets all outstanding technical requirements.

Kind regards,

Rudolf Kirchmair

Academic Editor

PLOS ONE

Additional Editor Comments (optional):

Reviewers' comments:

Reviewer's Responses to Questions

**Comments to the Author**

1. If the authors have adequately addressed your comments raised in a previous round of review and you feel that this manuscript is now acceptable for publication, you may indicate that here to bypass the “Comments to the Author” section, enter your conflict of interest statement in the “Confidential to Editor” section, and submit your "Accept" recommendation.

Reviewer #1: All comments have been addressed

2. Is the manuscript technically sound, and do the data support the conclusions?

Reviewer #1: Yes

3. Has the statistical analysis been performed appropriately and rigorously? 

Reviewer #1: Yes

4. Have the authors made all data underlying the findings in their manuscript fully available?

Reviewer #1: Yes

5. Is the manuscript presented in an intelligible fashion and written in standard English?

Reviewer #1: Yes

6. Review Comments to the Author

Reviewer #1: You have responded and replied appropriately to my review comment. I wish you luck in further developing your series study by collecting data on risk factors, severity of disease, and impact of treatment and other factors.

7. PLOS authors have the option to publish the peer review history of their article (what does this mean?). If published, this will include your full peer review and any attached files.

Reviewer #1: No

---

## [Editor Report · Acceptance letter]

26 Jul 2022

PONE-D-22-00911R1 

Risk of Peripheral Artery Occlusive Disease in Patients with Lower Leg Fracture Who Received Fixation and Non-fixation Treatments: A Population Cohort Study 

Dear Dr. Wu:

I'm pleased to inform you that your manuscript has been deemed suitable for publication in PLOS ONE. Congratulations! Your manuscript is now with our production department. 

Kind regards, 

on behalf of

Prof Rudolf Kirchmair 

Academic Editor

PLOS ONE